# Diagnosis of SARS-CoV-2 during the Pandemic by Multiplex RT-rPCR hCoV Test: Future Perspectives

**DOI:** 10.3390/pathogens11111378

**Published:** 2022-11-18

**Authors:** Alessio Danilo Inchingolo, Ciro Isacco Gargiulo, Giuseppina Malcangi, Anna Maria Ciocia, Assunta Patano, Daniela Azzollini, Fabio Piras, Giuseppe Barile, Vito Settanni, Antonio Mancini, Grazia Garofoli, Giulia Palmieri, Chiara Di Pede, Biagio Rapone, Angelo Michele Inchingolo, Megan Jones, Alberto Corriero, Nicola Brienza, Antonio Parisi, Angelica Bianco, Loredana Capozzi, Laura Del Sambro, Domenico Simone, Ioana Roxana Bordea, Gianluca Martino Tartaglia, Antonio Scarano, Felice Lorusso, Luigi Macchia, Giovanni Migliore, Van Hung Pham, Gianna Dipalma, Francesco Inchingolo

**Affiliations:** 1Department of Interdisciplinary Medicine, Section of Dental Medicine, University of Bari “Aldo Moro”, 70124 Bari, Italy; 2Department of Interdisciplinary Medicine—Intensive Care Unit Section, Aldo Moro University, 70121 Bari, Italy; 3Experimental Zooprophylactic Institute of Puglia and Basilicata, 71121 Foggia, Italy; 4Department of Oral Rehabilitation, Faculty of Dentistry, Iuliu Hațieganu University of Medicine and Pharmacy, 400012 Cluj-Napoca, Romania; 5UOC Maxillo-Facial Surgery and Dentistry, Department of Biomedical, Surgical and Dental Sciences, School of Dentistry, Fondazione IRCCS Ca Granda, Ospedale Maggiore Policlinico, University of Milan, 20100 Milan, Italy; 6Department of Innovative Technologies in Medicine and Dentistry, University of Chieti-Pescara, 66100 Chieti, Italy; 7Department of Emergency and Organ Transplantation (D.E.T.O.), University of Bari “Aldo Moro”, 70124 Bari, Italy; 8Interdisciplinary Department of Medicine, University Hospital of Bari, 70100 Bari, Italy; 9Department of Microbiology, Phan Chau Trinh University, Danang City 550000, Vietnam; 10International Institute of Gene and Immunology, Ho Chi Minh City 70000, Vietnam

**Keywords:** human pathogenic coronavirus, multiplex RT-rPCR, diagnostic accuracy, SARS-CoV-2, SARS-CoV, COVID-19, vaccines, antispike

## Abstract

Coronavirus disease 2019 (COVID-19), caused by severe acute respiratory syndrome coronavirus 2 (SARS-CoV-2), has rapidly become a significant threat to public health. However, among the Coronaviridae family members, there are other viruses that can also cause infections in humans. Among these, severe acute respiratory syndrome (SARS-CoV) and Middle East respiratory syndrome (MERS-CoV) have posed significant threats to human health in the past. Other human pathogenic coronaviruses have been identified, and they are known to cause respiratory diseases with manifestations ranging from mild to severe. In this study, we evaluated the performance of a multiplex RT-rPCR specific to seven human pathogenic coronaviruses in mainly detecting SARS-CoV-2 directly from nasopharyngeal swabs obtained from suspected COVID-19 infected patients, while simultaneously detecting different human pathogenic coronaviruses in case these were also present. We tested 1195 clinical samples suspected of COVID-19 infection. The assay identified that 69% of the samples tested positive for SARS-CoV-2 (1195), which was confirmed using another SARS-CoV-2 RT-PCR kit available in our laboratory. None of these clinical samples were positive for SARS-CoV, MERS-CoV or HCoV. This means that during the endemic phase of COVID-19, infection with other human pathogenic coronaviruses, even the common cold coronavirus (HCoV), is very uncommon. Our study also confirmed that the multiplex RT-rPCR is a sensitive assay for detecting SARS-CoV-2 regardless of differences among the variants. This multiplex RT-rPCR is also time- and cost-saving and very easy to apply in the diagnostic laboratory due to its simple procedure and its stability in storage after preparation. These features make the assay a valuable approach in screening procedures for the rapid detection of SARS-CoV-2 and other human pathogenic coronaviruses that could affect public health.

## 1. Introduction

To date, including the recently discovered SARS-CoV-2, there are seven coronaviruses that can infect humans. Human pathogenic coronaviruses belong to two genera, alpha coronavirus and beta coronavirus, among the four genera of the coronavirus sub-family: HCoV-229E and HCoV-NL63, and HCoV-HKU1, HCoV-OC43, SARS-CoV and MERS-CoV, respectively [1]. The human coronaviruses HCoV-229E, HCoV-NL63, HCoV-OC43 and HCoV-HKU1 cause only the common cold [2], whereas severe acute respiratory syndrome coronavirus (SARS-CoV) and Middle East respiratory syndrome coronavirus (MERS-CoV) cause relatively high mortality. They emerged in 2002 [3] and 2012 [4,5,6,7,8,9,10], respectively. The recently identified SARS-CoV-2 is currently causing a worldwide epidemic [11]. Based on the recent experience of the COVID-19 outbreak, which has spread rapidly across the globe and become a significant public health emergency of international concern, there is an increasingly urgent need for large-scale screening methods to make timely diagnoses of infections caused by human pathogenic coronaviruses from the family *Coronaviridae* [12]. Therefore, COVID-19 laboratory diagnostics are critical for containing the pandemic, slowing infections and allowing proper clinical care to take place [13,14,15,16,17]. There are several methods available for detecting human pathogenic coronaviruses.

The traditional method for detecting human pathogenic coronaviruses involves cell culture isolation of the virus from clinical specimens [18]. However, this method is feasible for only a small number of specialized laboratories and is time-consuming, making it inapplicable as a routine diagnostic method [19]. Among the modern methodologies, a next-generation sequencing approach offers the possibility of investigating new pathogens such as SARS-CoV-2. The sequencing of the entire genome of human pathogenic coronaviruses using NGS technology is a good approach, mainly because it contributes to identifying new variants of the virus and discovering unknown coronaviruses [20]. However, this technology requires sophisticated bioinformatic analysis and is expensive and time-consuming [21].

However, among the traditional or innovative methods, reverse transcriptase real-time PCR (RT-rPCR) technology is currently the most extensively used approach for identifying SARS-CoV-2 [22,23], and the MPL RT-rPCR represents a gold standard for rapid detection of human pathogenic coronaviruses [24]. Several diagnostic kits have been commercially developed as a result of the SARS-CoV-2 pandemic. However, most of them exclusively target the identification of SARS-CoV-2 [25]. They can provide presence/absence results using different viral targets or predict the SARS-CoV-2 variant based on the presence/absence of characteristic single-nucleotide variants (SNVs). Very few kits are commercially available that can simultaneously detect and identify all human coronaviruses including the current pandemic coronavirus SARS-CoV-2 [26,27]. This study aims to evaluate the ability of the rapid and sensitive diagnostic procedures for multiple detections of the Coronaviridae family patented by Pham et al., 2020 [28] to primarily detect SARS-CoV-2 as well as other human pathogenic coronaviruses (if they are present) in nasopharyngeal swabs collected from individuals with suspected COVID-19. In this study, we have introduced some modifications to the patent’s protocol to fit with our available materials. Additionally, we evaluated the stability of the multiplex RT-rPCR master mix conferred by the enzyme stabilizer that is rarely reported or evaluated in any commercial or home-made kits.

## 2. Materials and Methods

### 2.1. Sample Collection and RNA Isolation

The nasopharyngeal swabs were collected from suspected COVID-19 infected patients, between 1 July 2021 and 31 January 2022, from several hospitals in Apulia (Italy) and subsequently sent to Genetic and Molecular Epidemiology Laboratory of Experimental Zooprophylactic Institute of Apulia and Basilicata (IZSPB). No information about symptomatology and vaccination status was available.

Viral RNA was extracted from nasopharyngeal swabs in virus transport medium (COPAN’s Collection & Transport Kits for COVID-19). According to the manufacturer’s instructions, RNA extraction was performed from 200 µL of the virus transport medium using the QIAamp 96 Virus QIAcube HT Kit (Qiagen). Before RNA extraction, we added 20 µL of PEDV (Porcine Epidemic Diarrhea Virus), a Positive Control (PC) previously diluted in 200 µL of TE [28], which consists of the intact and inactive virus, to each sample. This PC was used as external control during the RNA extraction procedure. The extracted RNA was stored at −80 °C until the assay was performed.

### 2.2. Multiplex Reverse Transcriptase Real-Time PCR Assay

The University “Aldo MORO” of Bari, collaborating with Phan Chau Trinh University, Quang Nam province (Vietnam), developed a patent named Method and diagnostic kit for multiple detection of virus of the Coronaviridae family: SARS-CoV-2, SARS-CoV, HCoV and MERS-CoV. This patent was filed on 20 May 2020 in Italy (number: 102020000011701). It has also been extended to Europe (number: 20197796.4, filed on 23 September 2020), the United States (number: 17/034,407, filed on 28 September 2020), and Hong Kong (number: 22020018040.8, filed on 14 October 2020) [28]. This method combines RT-rPCR in a single-step approach which makes it possible to detect SARS-CoV-2, SARS-CoV, MERS-CoV and HCoV in a single response [28].

This study followed the above patented method. The names and the sequences of the primers and probes were previously described [28] and reported in Table 1.

For detection of SARS-CoV-2, three sets of primers and TaqMan probes targeted to N gene were used. For detection of SARS-CoV and MERS-CoV, two sets of primers and TaqMan probes targeted to E gene were used. For detection of common cold human coronavirus (HCoV), a set of primers and probes targeted to the replicase gene was used. To control the quality of the nasopharyngeal swab, a set of primers and probes targeted to the human RNAseP gene was used, and to control the quality of the RNA extraction step, a set of primers and probes targeted to the N gene of the PED virus was used.

Unlike Pham et al., 2020 [28], the SuperScript™ III Platinum™ One-Step qRT-PCR Kit (Invitrogen-ThermoFisher), following the recommendations of the manufacturer, was used to prepare the MPL RT-rPCR mix. Two MPL RT-rPCR master mixes, the MPL1 and the MPL2, were prepared as reported in Table 2.

The MPL1 was used to detect SARS-CoV-2 via the detection of N1, N2 and N3 from N gene of the virus, and to check for the existence of host epithelial cells in the samples via the detection of the human RNAse P gene from the host epithelial cells in the nasopharyngeal swabs. The MPL2 was utilized to detect both SARS-CoV and SARS-CoV-2 via the detection of its E gene, detect hCOV via the detection of its replicase gene and MERS-CoV via the detection of its upE gene. In addition, the MPL2 controlled the quality of the RNA extraction step via the detection of the N gene of porcine epidemic diarrhea virus (PED virus) extracted from the intact virus added to the sample before RNA extraction.

After preparation, the MPL1 and MPL2 were aliquoted into separated PCR tubes (for 2000 reactions each), then kept at −20 °C until used. Before using the MPL1 and MPL2 for the assay, the sensitivity of the MPL1 and MPL2 in the detection of each target gene was confirmed by testing with the 10× serial dilutions of the synthesized targeted oligo for SARS-CoV-2, SARS-CoV, MERS-CoV and HCoV. The MPL1 and MPL2 were accepted to be used in the assay when the LOD (limit of detection) reached 1 to 10 copies of the target oligo of SARS-CoV-2 and HCoV, 1–20 copies of the target oligo of MERS-CoV for each reaction volume (20 µL), as presented in Pham et al., 2020 [28].

To prepare the MPL RT-rPCR assay, the extracted RNA of the sample was added into one PCR tube containing the MPL1 and one PCR tube containing the MPL2; each PCR tube received 5 µL. For each run of the MPL RT-rPCR assay, 5 µL of the PC 1 containing 10–100 copies of the targeted oligo of SARS-CoV-2 was added to one MPL1 PCR tube, and 5 µL of the PC 2 containing 10–100 copies of the targeted oligo of SARS-CoV, MERS-CoV and HCoV was added to one MPL2 PCR tube. In addition, for each run of the MPL RT-rPCR assay, 5 µL of TE1X was added as the negative control (NC) to one MPL1 PCR tube, and to one MPL2 PCR tube. The thermal cycler profile consisted of 50 °C for 20 min and 95 °C for 5 min, followed by 45 cycles at 95 °C for 15 s and 60 °C for 50 s. The assay was conducted on the CFX Connect Real-Time PCR Detection System (Bio-Rad). Figure 1 shows the input of RNA extracted from the sample, the PC and the NC (TE1X) into the MPL1 and MPL2.

After the MPL RT-rPCR was run, the interpretation of the results was based on the amplification curves of 4 fluorescent channels in MPL1 and MPL2, as the instructions presented in Table 3 show.

### 2.3. Confirmation of the Multiplex RT-rPCR in the Detection of SARS-CoV-2

The GSD NovaType II SARS-CoV-2 and GSD NovaType III SARS-CoV-2 kits (Eurofins Technologies) were used to confirm SARS-CoV-2 positive and negative samples. Both the real-time RT-PCR assays are based on identifying relevant mutation associated with SARS-CoV-2 variants. The study used two different tests because the GSD NovaType III SARS-CoV-2 kit was unavailable when the study was started. The reaction was performed following the manufacturer’s instructions. The thermal cycler profile consisted of 50 °C for 10 min and 95 °C for 3 min, followed by 40 cycles at 95 °C for 10 s and 60 °C for 30 s. The reaction was also conducted on the CFX Connect Real-Time PCR Detection System (Bio-Rad, Hercules, CA, USA) and analyzed by CFX Maestro ^TM^ Software 2.3 (Bio-Rad, Hercules, CA, USA).

### 2.4. Enzyme Stabilizer Validation

As reported in Table 2, the multiplex RT-rPCR master mix also consists of a reagent named “Enzyme Stabilizer”, which stabilizes the enzyme reverse transcriptase over time. To validate the role of this reagent, we tested 321 samples at time zero, i.e., when the samples were collected in our laboratory, and after six months, to evaluate the performance of the assay and compare it over time based on Ct value.

### 2.5. Statistical Analysis

To compare the Ct value obtained during independent multiplex RT-rPCR assay, an unpaired t-test was performed using GraphPad Prism version 9.3.1 (Inc., San Diego, CA, USA).

## 3. Results

Over six months, we collected 1195 nasopharyngeal swabs, which were sent to us randomly from six hospitals in Puglia: 43% (517/1195) from ASL Lecce; 26% (306/1195) from ASL Brindisi; 18% (213/1195) from ASL Taranto; 3% (38/1195) from I.R.C.C.S. Saverio De Bellis—Ente Ospedaliero; 2% (28/1195) from ASL Bari; and 0.75% (9/1195) from Ospedale Generale Regionale F. Miulli. The remaining 7% (84/1195) of the samples were collected from the drive-in service of the Experimental Zooprophylactic Institute of Apulia and Basilicata in Foggia. Among the samples 51% (613/1195) were from males, and 49% (582/1195) were from females, and the patients’ ages ranged from 0 to 96 years. No information about vaccination status or clinical manifestations was available.

The RNA purified from 1195 nasopharyngeal specimens was tested with the multiplex RT-rPCR assay. Overall, the assays showed a percent positive of 69% (835/1195) for SARS-CoV-2. Amplification curves obtained from the RT-rPCR were interpreted according to Pham et al., 2020 [28], with example cases presented in Figure 2. The Ct value obtained for each target is shown in Figure 3.

In all samples tested, including negative SARS-CoV-2 samples, we revealed the amplification curves of RNaseP in the MPL1 and PEDV nucleocapsid (N) in the MPL2, used as an internal control of specimens and a positive control for the extraction process, respectively. In all of the analyzed samples, positivity was not detected for the remaining targets of the MPL2, which allows for detection of the infection caused by MERS-CoV, SARS-CoV or HCoVs. No amplification curve was detected in the negative control. The comparative diagnosis test performed using the additional kit for SARS-CoV-2 confirmed all SARS-CoV-2-positive specimens (*n* = 835). In addition, owing to the characteristics of the confirmatory test, it was possible to define the probable variants of SARS-CoV-2 (Appendix A), thus attesting to the good performance of the multiplex RT-rPCR and the ability of the MPL RT-rPCR to detect all samples positive for SARS-CoV-2 regardless of differences in the variants. Appendix A also presents the different variants of SARS-CoV-2 predicted by GSD NovaType II SARS-CoV-2/GSD NovaType III SARS-CoV-2 kits and subsequently confirmed by whole genome sequencing (Pangolin Lineage). Appendix A illustrates that the samples detected to be positive for SARS-CoV-2 by the MPL RT-rPCR were among different variants and none tested negative with the MPL RT-rPCR assay.

In order to evaluate the stability of the MPL1 and MPL2 over time, the multiplex RT-rPCR assay was repeated on a total of 321 SARS-CoV-2-positive RNA extracts, stored at −80 °C, six months after the first diagnosis. The assay confirmed virus positivity, providing Ct values similar to those obtained during the first real-time session (Figure 4).

In agreement with this result, no statistical significance was found when comparing Ct values between the two sessions (*p* > 0.05).

## 4. Discussion

Under the pressure of the COVID-19 pandemic around the world, different kits based on different approaches have been produced, all aimed at the rapid detection of SARS-CoV-2. As strongly proposed by some authors, the objective is to develop and implement multi-specific combination screening procedures that enable scientists to target bacterial, fungal, or virus-specific diseases in a short time to produce world-scale qualitatively tactical therapeutic activities. Moreover, shortening the time needed will aid in limiting both the economic and social impacts of pandemic outbreaks [29,30,31]. Among the available methods, the RT-rPCR represents a reliable approach to diagnosing SARS-CoV-2 with high sensitivity and specificity [32].

The purpose of this investigation was to test the simultaneous identification of *Coronaviridae* family viruses in clinical specimens using a multiplex RT-rPCR assay, which is capable of identifying, with a single reaction, not only SARS-CoV-2 but also six other human pathogenic coronaviruses for public health interest. This represents a novelty because no other comparable kits are commercially available, nor have they been described in the literature. As is generally known, coronaviruses have been responsible for three major outbreaks since the beginning of the 21st century [33]; thus, it could be helpful for public health surveillance to have a cheap and convenient test that could help to identify not only SARS-CoV-2, but also other human pathogenic coronaviruses that could in the future pose critical threats to the world’s population. Epidemiologic surveillance requires appropriate tools and knowledge so that it cannot fail a second time in the future as, unfortunately, it did for the current ongoing pandemic. The multiplex real-time PCR kit test used in this study has several advantages [28]: (1) It incorporates RT-rPCR technology into a one-step method. (2) Using this unique RT-rPCR-based technology, it in fact detects four viral targets in a single analysis. The new test enables customers to recognize all coronaviruses known to be dangerous pathogens to humans, including hCoVs, which cause the flu, SARS-CoV, which causes SARS, SARS-CoV-2, which causes COVID-19, and MERS-CoV, which causes MERS [34,35,36]. (3) The assay described here is very rapid. In fact, it took less than two hours to test 96 nasopharyngeal samples (the assay can also be carried out on other biological matrices). According to these considerations, this test should not be underestimated, and it provides a considerable advantage for those involved in epidemiological surveillance. (4) A verification process was established to indicate the existence of possible external and internal contamination in order to confirm the occurrence of negative results, and also the high level of sensitivity with the amplification phase via the DNA positive control. (5) A check-step was established to prevent false negatives by using the PEDV-CoV and the RP gene as internal controls. (6) The MPL-rPCR is extremely sensitive and can identify both non-cultivatable and neutralized viruses in antigen-antibody complexes [37]. The RT-rPCR approach is still a robust and reliable way to confirm the presence of COVID-19 infection, mainly when used in conjunction with objective clinical screenings and diagnostic testing modalities [38]. Moreover, due to the high level of sensitivity and specific targeting used in the evaluating process, already reported in Pham et al., 2020 [28], this recent kit-diagnostic system may be extremely helpful in achieving the highest possible level of consistency and accuracy in identifying the different coronavirus strains such as SARS-CoV, SARS-CoV-2, HCoV, and MERS-CoV [39,40]. In addition, the use of the stabilizing enzyme simplifies the assay set-up procedures and allows for a significant reduction in run times. In fact, it is possible to prepare a large quantity of the two master mixes, which can be dispensed in smaller volumes and stored at −20 °C. As we have shown in this study, the performance of the reaction mixes is not affected by thawing and freezing. However, it is always good practice to prepare reaction volumes suitable for the laboratory’s needs (i.e., aliquots for 100 reactions). This result represents a novelty. While it is true that there are commercially available kits consisting of master mixes, these generally do not include the primer sets, which are more easily degraded. Additionally, in our study we evaluated the possibility of applying the diagnostic kit as an open system, adapting it to the RNA extraction kit in use at our laboratory and setting up the two multiplex RT-rPCR mixtures using a different reverse transcriptase enzyme. These modifications were discussed and approved by the authors who patented the kit. Our results showed the good performance of the assay, as demonstrated by the authors [28]. These results were supported by the confirmatory assay used in our study, which used a different target but produced comparable Ct values. In addition, since the confirmatory assay is a test that allows us to predict the variant of SARS-CoV-2, we can affirm that the multiplex RT-rPCR assay allows diagnosis of the infection despite the virus’s genome mutations. This aspect is relevant, because many SARS-CoV-2 mutations can occur throughout the genome, especially at the spike gene; however, several mutations also involve the N gene [41]. Since the target of this assay involves three different loci, it does not affect the sensitivity of the multiplex RT-rPCR assay. This finding was confirmed by the detection of five variants of SARS-CoV-2 (alpha, gamma, delta, lambda and omicron) by the GSD NovaType II SARS-CoV-2/GSD NovaType III SARS-CoV-2 and the NGS among the SARS-CoV-2-positive samples detected by the MPL RT-rPCR. Again, we can say that the MPL-RT-rPCR is sensitive to all cases of SARS-CoV-2 that exist in a sample, regardless of variant.

Although 69% of our tested samples were positive for SARS-CoV-2 infection, an additional advantage of the assay tested is its ability to simultaneously identify other members of the *Coronaviridae* family which are capable of causing infections in humans, because this MPL RT-rPCR could detect not only SARS-CoV-2 but also other human pathogenic coronaviruses like SARS-CoV, MERS-CoV and hCoV. However, in this study, none of these viruses were detected. From this finding, we can conclude that among the suspected COVID-19 infected patients whose samples were taken during the COVID-19 epidemic, infection or coinfection with other human pathogenic coronaviruses was very uncommon.

## 5. Conclusions

In conclusion, the assay demonstrates excellent performance, and the reagent cost is relatively low, making this method ideal for large-scale population screening. Although, in this study, no other human pathogenic coronaviruses apart from SARS-CoV-2 were identified in the samples that were brought to our laboratory, this is probably because SARS-CoV-2 was the main human pathogenic coronavirus present in the territory studied during the time of our investigation. Nevertheless, we tested the assay’s ability to simultaneously identify the presence of the targets by assaying a unique mixture of positive controls for both multiplex RT-rPCR master mixes; tests with the same aim were performed and referenced in the previous study and in the patent. The assay is easily performed and does not require specialized personnel. Contrary to other assays available on the market, it can simultaneously identify the presence of different potentially lethal human pathogenic viruses of the *Coronaviridae* family [42]. The advantage of providing two multiplex RT-rPCR master mixes is no less important, as it remains stable over time thanks to the addition of the enzyme stabilizer. These advantages make the assay rapid and easy to use.

## 6. Patents

The following is a patent resulting from the work reported in this manuscript: 102020000011701. Name: Università Degli Studi di Bari Aldo Moro (80%). Phan Chau Trinh University (20%). Inventors’ names: Andrea Ballini, Francesco Inchingolo, et al. Title: “Metodo e kit diagnostico per l’individuazione multipla di virus della famiglia Coronaviridae: SARS-CoV2, SARS-CoV, HCoV e MERS-CoV”. In addition, it has also been extended to Europe (number: 20197796.4, filed on 23 September 2020), the United States (number: 17/034,407, filed on 28 September 2020) and Hong Kong (number: 22020018040.8, filed on 14 October 2020).

## Figures and Tables

**Figure 1 pathogens-11-01378-f001:**
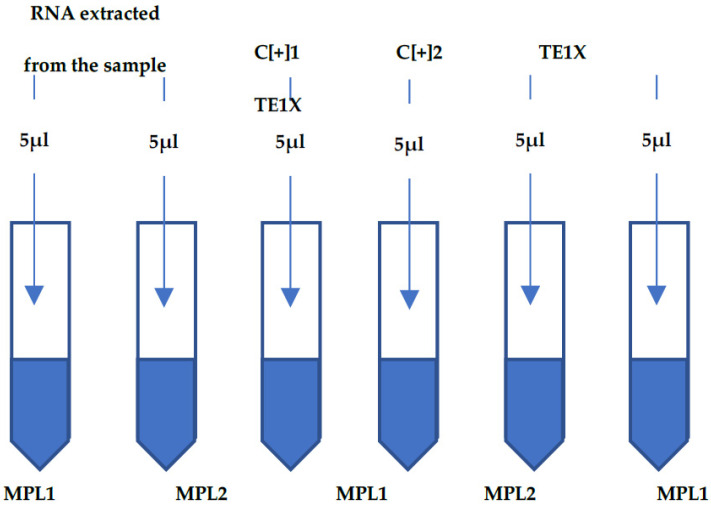
Shows the input of RNA extracted from the sample, the PC and the NC into the MPL1 and MPL2 to run the MPL RT-rPCR. The PC and NC were added to the MPL1 and MPL2 for each run of the assay, not for each sample.

**Figure 2 pathogens-11-01378-f002:**
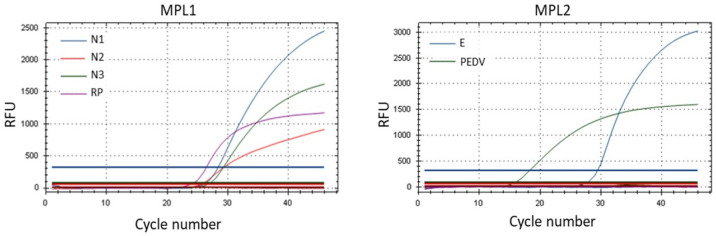
Example of amplification curve for a positive SARS-CoV-2 sample. Multiple reaction curves indicate amplification of the positive clinical sample using two multiplex RT-rPCR master mixes. N1–3: SARS-CoV-2 nucleocapsid gene; RP: RNaseP gene; E: SARS-CoV-2 envelope gene; PEDV: porcine epidemic diarrhea virus.

**Figure 3 pathogens-11-01378-f003:**
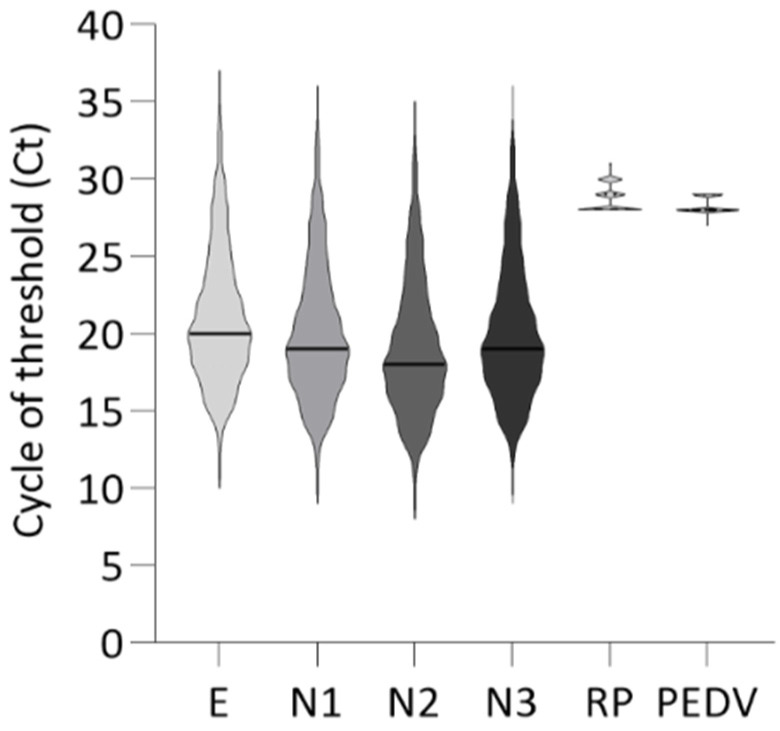
Violin plot of the cycle threshold (Ct) value of multiplex RT-rPCR. The violin plot shows the cycle threshold (Ct) values obtained for each positive SARS-CoV-2 sample tested (*n* = 835). For each plot, the continuous line indicates the median value: E = 20; N1 = 19; N2 = 18; N3 = 19; RP = 28; PEDV = 28. No amplification curve was obtained for HCoV-HKU-1-PR or upE_TqPR. E = envelope gene; N1-N3 = nucleocapsid gene; RP = RNaseP gene; PEDV = porcine epidemic diarrhea virus. The data were plotted using GraphPad Prism 9.3.1.

**Figure 4 pathogens-11-01378-f004:**
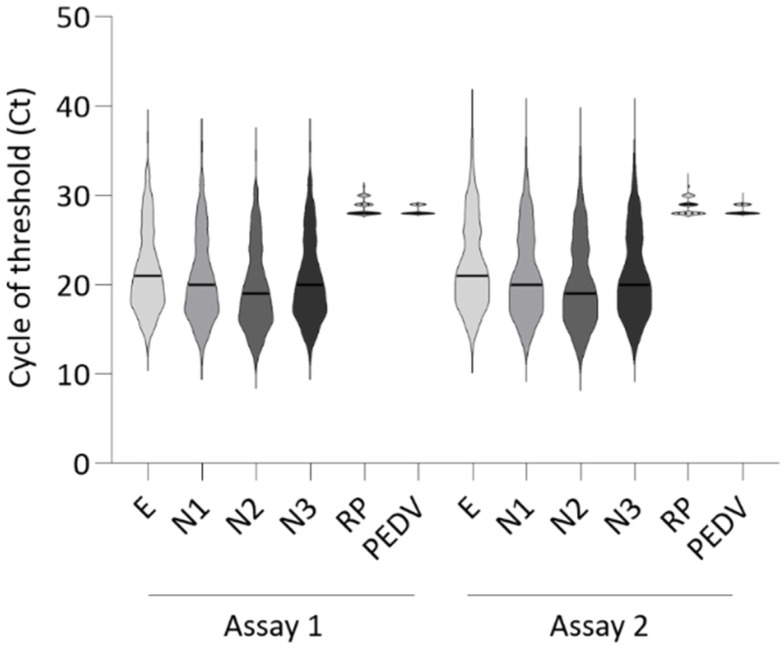
Comparison of cycle threshold (Ct) values of two multiplex RT-rPCR sessions performed at different times. The violin plot shows the cycle threshold (Ct) values obtained during two different assays performed on 321 positive SARS-CoV-2 samples at time zero (Assay 1) and after six months (Assay 2). The plot shows that the Ct value is very similar between the two independently performed assays. This result confirmed the stability of multiplex RT-rPCR master mix conferred by Enzyme Stabilizer. The continuous line inside the plot indicates the median value. Assay 1: E = 21; N1 = 20; N2 = 19; N3 = 20; RP = 28; PEDV = 28. Assay 2: E = 21; N1 = 20; N2 = 19; N3 = 20; RP = 28; PEDV = 28. No significant difference (*p* < 0.05) was revealed by t-test. E = envelope gene; N1–N3 = nucleocapsid gene; RP = RNaseP gene; PEDV = porcine epidemic diarrhea virus. The data were plotted using GraphPad Prism 9.3.1.

**Table 1 pathogens-11-01378-t001:** Nucleotide sequence of primers used to prepare two multiplex RT-rPCR master mixes (MPL1 and MPL2). N1–3: nucleocapsid of SARS-CoV-2; RNAseP: human reference; E: envelope of SARS-CoV-2; UpE: envelope of SARS-CoV; UpE: envelope of MERS-CoV; N: nucleocapsid of PEDV; RdRp: RNA-dependent RNA polymerase of hCoVs.

Oligo Name	Target	Mix	Sequence 5′–3′
2019-nCoV_N1-F	N1	MPL1	GACCCCAAAATCAGCGAAT
2019-nCoV_N1-R			TCTGGTTACTGCCAGTTGAATCTG
2019-nCoV_N1-Probe			FAM-ACCCCGCATTCAGTTTGGTGGACC-BHQ1
2019-nCoV_N2-F	N2		TTACAAACATTGGCCGCAAA
2019-nCoV_N2-R			GCGCGACATTCCGAAGAA
2019-nCoV_N2-Probe			TexasRED-ACAATTTTGCCCCCAGCGCTTCAG-BHQ2
2019-nCoV_N3-F	N3		GGGAGCCTTGAATACACCAAAA
2019-nCoV_N3-R			TGTAGCACGATTGCAGCATTG
2019-nCoV_N3-Probe			HEX-AYCACATTGGCACCCGCAATCCTG-BHQ1
RP-F	RNAseP		AGATTTGGACCTGCGAGCG
RP-R			GAGCGGCTGTCTCCA
RP-Probe			CY5-TTCTGACCTGAAGGCTCTGCGCG-BHQ3
E_Sarbeco_F1	E	MPL2	ACAGGTACGTTAATAGTTAATAGCGT
E_Sarbeco_R2			ATATTGCAGCAGTACGCACACA
E_Sarbeco_Probe			FAM-ACACTAGCCATCCTTACTGCGCTTCG-BHQ1
upE_TqF	UpE		GCAACGCGCGATTCAGTT
upE_tqR			GCCTCTACACGGGACCCATA
upE_TqProbe			FAM-CTCTTCACATAATCGCCCCGACGTCG-BHQ2
PEDV-NF	N		GCGCAAAGACTGAACCCACTA
PEDV-NR			TTGCCTCTGTTGTTACTTGGAGAT
PEDV-Probe			HEX-TGTTGCCATTGCCACGACTCCTGC-BHQ1
HCoV-HKU-1-F	RdRp		CCTTGCGAATGAATGTGCT
HCoV-HKU-1-R			TTGCATCACCACTGCTAGTACCAC
HCoV-HKU-1-Probe			CY5-TGTGTGGCGGTTGCTATTATGTTAAGCCTG-BHQ3

**Table 2 pathogens-11-01378-t002:** Volumes of reagents for reactions.

Reagent	MPL1	MPL2
H_2_O (RNAse free)	15 μL	15 μL
2× Buffer	10 μL	10 μL
SuperScript^TM^ III RT/Platinum^TM^ Taq High Fidelity Enzyme mix	0.4 μL	0.4 μL
Enzyme stabilizer	1 μL	1 μL
2019-nCoV_N1-F	10 pm	-
2019-nCoV_N1-R	10 pm	-
2019-nCoV_N1-Probe	5 pm	-
2019-nCoV_N2-F	10 pm	-
2019-nCoV_N2-R	10 pm	-
2019-nCoV_N2-Probe	5 pm	-
2019-nCoV_N3-F	10 pm	-
2019-nCoV_N3-R	10 pm	-
2019-nCoV_N3-Probe	5 pm	-
RP-F	10 pm	-
RP-R	10 pm	-
RP-Probe	5 pm	-
E_Sarbeco_F1	-	10 pm
E_Sarbeco_R2	-	10 pm
E_Sarbeco_Probe	-	5 pm
upE_TqF	-	10 pm
upE_tqR	-	10 pm
upE_TqProbe	-	5 pm
PEDV-NF	-	10 pm
PEDV-NR	-	10 pm
PEDV-Probe	-	5 pm
HCoV-HKU-1-F	-	2 pm
HCoV-HKU-1-R	-	2 pm
HCoV-HKU-1-Probe	-	5 pm

**Table 3 pathogens-11-01378-t003:** Instructions for the interpretation of the results based on the amplification curves of 4 fluorescent channels that exist in the MPL1 and MPL2.

MPL1				MPL2				Interpretation
FAM	HEX	Texas-RED	CY5	FAM	HEX	Texas-RED	CY5	
+	+	+	+ ^1^	+ ^2^	+	−	−	SARS-CoV-2
−	−	−	+	−	+	−	−	No coronaviral pathogens detected
−	−	−	−	−	+	−	−	Sample contains no epithelial cells
−	−	−	−	−	−	−	−	The RNA extraction step was failed
−	−	−	+	+	+	−		SARS-CoV
−	−	−	+	−	+	+	−	MERS-COV
−	−	−	+	−	+	−	+	hCOV

*Note*: ^1^ If SARS-CoV-2 in the sample is sufficiently high, then the amplification of RNAse P gene may be completed and the CY5 amplification signal in the MPL1 may be weak or absent. ^2^ The primers and probes specific for SARS-CoV used in this study are also specific for SARS-CoV-2. In silico analysis was performed for testing the specificity of primers by Sanger sequencing of PCR fragment obtained.

## Data Availability

All experimental data to support the findings of this study are available from the corresponding author upon request.

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
