# Peer review of "Diagnosis of SARS-CoV-2 during the Pandemic by Multiplex RT-rPCR hCoV Test: Future Perspectives"

_pathogens, 2022, doi:10.3390/pathogens11111378_

Round 1

Reviewer 1 Report

 This manuscript is evaluated the performance of a multiplex RT-rPCR specific for 7 human pathogenic coronaviruses. It is significant in screening procedures for rapid detection of SARS-CoV-2 and other human 63 pathogenic corona-viruses relevant to public health. Several questions need to be further answered from the author:

1) the authors tested 1195 clinical samples suspected of COVID-19, None of case among these clinical samples 56 were positive with SARS-CoV, MERS-CoV and HCoV. Then the anthor concluded that "during the endemic of  COVID-19, the in-fection of the other human pathogenic coronaviruses, even the common cold coronavirus (HCoV), is very uncommon." I suggested that in the study, they need expand the diversity of samples, and take samples from patients suspected of multiple infections to prove this conclusion. Because, the present result is influenced by limitations of the samples.

2)As mentioned in the manuscript, the results of the assay were confirmed by using another SARS-CoV-2 RT-PCR kit available in author's laboratory, the positive samples were also use NGS to obtain the sequence information of the virus. Dose the confirmatory test use the standard method? If the confirmatory test is not the standard method, it is very hard to avoid the false negative.

3)In addition, in the Enzyme Stabilizer validation experiment, all the regents are stored at - 80 ? almost all the regents are stabled in -80°C, I think that is not novel.

4)At the same , the test time is 2h is not novel. many test reagents are tested within 1h, some even within 30 minutes.

Author Response

Reviewer 1 

  • the authors tested 1195 clinical samples suspected of COVID-19, None of case among these clinical samples 56 were positive with SARS-CoV, MERS-CoV and HCoV. Then the anthor concluded that "during the endemic of  COVID-19, the in-fection of the other human pathogenic coronaviruses, even the common cold coronavirus (HCoV), is very uncommon." I suggested that in the study, they need expand the diversity of samples, and take samples from patients suspected of multiple infections to prove this conclusion. Because, the present result is influenced by limitations of the samples.

This is an interesting observation. However, at the moment, this study cannot be expanded because we cannot access more clinical samples and for the necessary documentation to be initiated and for willingness to process more clinical samples to be given by the Local Health Authorities, a long time is required. However, we believe it is important to share with the scientific community the results obtained by exploiting an assay that could facilitate screening investigations in diagnostic laboratories.

  • As mentioned in the manuscript, the results of the assay were confirmed by using another SARS-CoV-2 RT-PCR kit available in author's laboratory, the positive samples were also use NGS to obtain the sequence information of the virus. Dose the confirmatory test use the standard method? If the confirmatory test is not the standard method, it is very hard to avoid the false negative.

Confirmatory testing and whole viral genome sequencing were two additional methods to highlight the assay tested's efficacy. Since this is the first time the assay has been performed on clinical specimens, we thought it would be helpful to confirm the assay and verify that the numerous mutations found in the virus genome do not affect the efficacy of the assay. Certainly, it will not be necessary to perform the assays with confirmatory testing later.

  • In addition, in the Enzyme Stabilizer validation experiment, all the regents are stored at - 80 ? almost all the regents are stabled in -80°C, I think that is not novel.

The reagents are stored at -20°C, but apart from the freezing temperature, the advantage is to have a ready-made reaction mix to which only the RNA of the sample to be assayed needs to be added. This represents a novelty because kits on the market provide a master mix to which the primers, templating (DNA/RNA) and possibly water must be added. The use of the stabilizer allows precisely stabilizing the reagents in the master mix, as described in the text, especially the primers that degenerate in the freezing and thawing phases.

  • At the same, the test time is 2h is not novel. many test reagents are tested within 1h, some even within 30 minutes.

The advantage of the test is not only the speed of the execution but also the possibility of simultaneously testing multiple viruses of the Coronaviridae family. No such assay exists commercially, so as mentioned earlier, it is important to share these results.

Reviewer 2 Report

The manuscript is very elaborative and includes a range of results. However, I have few doubts which need to be addressed before publication:

1) What is the difference between the rRT-qPCR and multiplex RT-rPCR? How cost effective is multiplex RT-rPCR? 2) How did authors decide each of this primer and whether they designed it?  3) Did authors use other diagnostic tests to compare the test results?

Author Response

  • What is the difference between the rRT-qPCR and multiplex RT-rPCR? How cost effective is multiplex RT-rPCR?

The use of the term multiplex RT-rPCR  is related to the simultaneous identification of multiple targets in a precise multiplex reaction. When only one target is identified with a real-time assay, we define RT-rPCR. In contrast, RT-qPCR means an RNA transcript that quantifies the amount of cDNA (for the hCoVs), but that was not the aim of the study.

Regarding the costs, excluding primers and probes that can be used for a much larger number of reactions than kits/reagents characterized by a finite number of reactions, we estimated that the cost per reaction amounts to about 7 euros (7.02 dollars), including all steps from RNA extraction to test execution.

  • How did authors decide each of this primer and whether they designed it?

Primers and probes were chosen based on a survey of published data in the literature and organizations such as Centers for Disease Control and Prevention (CDC) and World Health Organization (WHO), as described in Pham et al., 2020. TaqMan probe sequences were chosen to be labeled with different fluorophores to make the assay a multiplex RT-rPCR.

  • Did authors use other diagnostic tests to compare the test results?

Certainly, the results obtained in this work were compared with those obtained from GSD NovaType II SARS-CoV-2 and GSD NovaType III SARS-CoV-2 kits, as described in the manuscript. In addition, in the test validation phase, random samples and controls were also assayed for the presence/absence of all targets with single RT-rPCR sessions (no in multiplex). The controls were always positive, confirming the assay performed in multiplex RT-rPCR and providing evidence that the assay has no inhibition when performing simultaneous target identification. However, since this is an optimization step in the reaction, we did not find it necessary to include it in the final draft of the manuscript.

Reviewer 3 Report

The authors reported a multiplex RT-rPCR specific for 7 human pathogenic coronaviruses. This is an important work in the diagnostic laboratory and the success is to be congratulated - the optimization of multiplex qPCR is hard, not to mention one-step qRT-PCR!

Some minor comments -

1. The assay may not be as well-validated on the non SARS-CoV-2 coronaviruses (Ln 366-367). The authors should make it clear to the readers that they need to perform local validation with relevant samples should they want to use it for own research or diagnosis. 

2. The authors could consider mixing of samples to test the co-detection case. In fact, co-infection by two human coronaviruses has been reported in the literature. 

3. Some wording would benefit from revising for a more factual and objective presentation -

- Line 54: The assay identified 69% of SARS-CoV-2 positive samples (I think you mean 69% of the 1,195 samples tested were positive, otherwise you are saying it has 69% sensitivity in the abstract!)

- Line 302: The purpose of the investigation is to highlight the existence of a multiplex RT-rPCR assay developed from Pham et al (it may be the authors' real intention, but it is not really a justifiable reason to do research to highlight the existence of another paper)

4. Table 4 occupies a lot of space without giving a lot of information. Consider a relevant sequence alignment showing SARS-CoV-2 sequence variation at the primed and probed regions, with representative strains from each clade (and noting the number of samples detected next to sequence from that sequence). 

Author Response

Reviewer 2 

  1. The assay may not be as well-validated on the non SARS-CoV-2 coronaviruses (Ln 366-367). The authors should make it clear to the readers that they need to perform local validation with relevant samples should they want to use it for own research or diagnosis.

We did internal testing to verify the ability of the assay to identify the simultaneous presence of the targets of interest by mixing positive controls. This allowed us to verify the assay's performance and only then was the assay tested on clinical samples. For greater clarity, we have reported this in the text. 

  1. The authors could consider mixing of samples to test the co-detection case. In fact, co-infection by two human coronaviruses has been reported in the literature.

A simulation of coinfection was carried out in the validation phase of the test. Similar to the PEDV control, a positive sample for more than one target virus was reproduced (by mixing the test positive control samples). The real-time PCR assay was performed to verify the simultaneous detection of targets (viruses) and, thus, the performance. These results have not been reported in the text as the objective is the exclusive use of clinical samples to make the working conditions more similar to those for which the study itself is proposed.

  1. Some wording would benefit from revising for a more factual and objective presentation -

- Line 54: The assay identified 69% of SARS-CoV-2 positive samples (I think you mean 69% of the 1,195 samples tested were positive, otherwise you are saying it has 69% sensitivity in the abstract!)

- Line 302: The purpose of the investigation is to highlight the existence of a multiplex RT-rPCR assay developed from Pham et al (it may be the authors' real intention, but it is not really a justifiable reason to do research to highlight the existence of another paper)

  1. Table 4 occupies a lot of space without giving a lot of information. Consider a relevant sequence alignment showing SARS-CoV-2 sequence variation at the primed and probed regions, with representative strains from each clade (and noting the number of samples detected next to sequence from that sequence). 
We modified the text and moved table 4 as a supplementary table.